# Functional and Molecular Changes in the Prefrontal Cortex of the Chronic Mild Stress Rat Model of Depression and Modulation by Acute Ketamine

**DOI:** 10.3390/ijms241310814

**Published:** 2023-06-28

**Authors:** Jessica Mingardi, Elona Ndoj, Tiziana Bonifacino, Paulina Misztak, Matteo Bertoli, Luca La Via, Carola Torazza, Isabella Russo, Marco Milanese, Giambattista Bonanno, Maurizio Popoli, Alessandro Barbon, Laura Musazzi

**Affiliations:** 1School of Medicine and Surgery, University of Milano-Bicocca, 20900 Monza, Italy; jessica.mingardi@unimib.it (J.M.); paulina.misztak@unimib.it (P.M.); 2Department of Molecular and Translational Medicine, University of Brescia, 25121 Brescia, Italy; e.ndoj@unibs.it (E.N.); m.bertoli029@studenti.unibs.it (M.B.); luca.lavia@unibs.it (L.L.V.); isabella.russo@unibs.it (I.R.); alessandro.barbon@unibs.it (A.B.); 3Department of Pharmacy, Unit of Pharmacology and Toxicology, University of Genoa, 16148 Genoa, Italy; tiziana.bonifacino@unige.it (T.B.); carola.torazza@unige.it (C.T.); marco.milanese@unige.it (M.M.); giambattista.bonanno@unige.it (G.B.); 4Inter-University Center for the Promotion of the 3Rs Principles in Teaching & Research (Centro 3R), 56122 Pisa, Italy; 5Genetics Unit, IRCCS Istituto Centro S. Giovanni di Dio Fatebenefratelli, 25125 Brescia, Italy; 6IRCCS Ospedale Policlinico San Martino, 16132 Genoa, Italy; 7Dipartimento di Scienze Farmaceutiche, Università Degli Studi di Milano, 20133 Milano, Italy; maurizio.popoli@unimi.it

**Keywords:** prefrontal cortex, chronic mild stress, ketamine, animal model, antidepressant, molecular mechanism, RNA editing

## Abstract

Stress is a primary risk factor in the onset of neuropsychiatric disorders, including major depressive disorder (MDD). We have previously used the chronic mild stress (CMS) model of depression in male rats to show that CMS induces morphological, functional, and molecular changes in the hippocampus of vulnerable animals, the majority of which were recovered using acute subanesthetic ketamine in just 24 h. Here, we focused our attention on the medial prefrontal cortex (mPFC), a brain area regulating emotional and cognitive functions, and asked whether vulnerability/resilience to CMS and ketamine antidepressant effects were associated with molecular and functional changes in the mPFC of rats. We found that most alterations induced by CMS in the mPFC were selectively observed in stress-vulnerable animals and were rescued by acute subanesthetic ketamine, while others were found only in resilient animals or were induced by ketamine treatment. Importantly, only a few of these modifications were also previously demonstrated in the hippocampus, while most are specific to mPFC. Overall, our results suggest that acute antidepressant ketamine rescues brain-area-specific glutamatergic changes induced by chronic stress.

## 1. Introduction

Major depressive disorder (MDD) is among the most debilitating disorders in the world, dramatically impacting healthcare systems. Genetic and environmental factors are recognized to contribute to the pathogenesis of MDD, with behavioral stress being one of the major risk factors [1,2]. Indeed, exposure to traumatic or chronic stress can induce maladaptive consequences in stress-vulnerable subjects that may ultimately increase the risk of neuropsychiatric disorders, including MDD [2,3,4]. Cortico-limbic glutamatergic brain regions, including the hippocampus, prefrontal cortex, and amygdala, were reported to undergo structural, functional, and connectivity alterations both in MDD patients and in stress-based animal models of depression [3,5,6,7,8].

For decades, the available pharmacological treatments for MDD have been limited to a few classes of drugs that, although effective in most patients, are limited by the onset of action delayed by several weeks and the lack of efficacy in up to a third of patients [9]. In this context, ketamine, a dissociative anesthetic and *N*-methyl-D-aspartate (NMDA) receptor antagonist, has recently gained attention for its rapid (within hours) and sustained (up to several days) antidepressant effect at a low subanesthetic dose in treatment-resistant MDD patients [10,11,12,13]. In 2019, this finding led to the approval by both the Food and Drug Administration (FDA) in the US and the European Medicine Agency (EMA) in Europe of the intranasal administration of the (S) enantiomer of ketamine in combination with a newly initiated oral antidepressant for the management of treatment-resistant depression [14].

In recent years, our group has extensively studied the hippocampal molecular underpinnings of the rapid antidepressant effect of ketamine using the chronic mild stress (CMS) animal model of depression in male rats [15,16,17]. We have demonstrated that CMS induces functional, molecular, and structural alterations selectively in the hippocampus of stress-vulnerable animals and that acute subanesthetic ketamine rescues most of these changes in just 24 h [15,16,17,18]. In particular, we have shown that vulnerability to CMS is characterized by the (i) hyperactivity of the hypothalamic–pituitary–adrenal (HPA) axis; (ii) an imbalance of excitatory/inhibitory neurotransmission, as shown by impairments of presynaptic glutamate and γ-aminobutyric acid (GABA) release; (iii) shortening and simplification of CA3 apical dendrites together with a reduction in brain-derived neurotrophic factor (BDNF) mRNA expression and dendritic trafficking; (iv) changes in the expression and activation of the main NMDA and α-amino-3-hydroxy-5-methyl-4-isoxazolepropionic acid (AMPA) receptor subunits; (v) decrease in the expression of glutamate metabotropic receptor mGluR2; and (vi) altered expression of brain-specific microRNAs and, in particular, lower levels of miR-9-5p and miR-135-5p. Notably, most of these alterations were restored by acute treatment with ketamine in just 24 h.

The present study focused on the medial prefrontal cortex (mPFC), a brain area involved in both cognitive and emotional deficits induced by stress [1,2,19]. We investigated whether CMS-induced molecular and functional changes in the mPFC of rats were associated with vulnerability/resilience to chronic stress, including alterations of presynaptic glutamate/GABA release, ionotropic and metabotropic glutamate receptor expression and activation, and post-synaptic plasticity. Moreover, we assessed if acute ketamine exerted a restorative effect on these changes.

## 2. Results

### 2.1. Chronic Mild Stress Increases the Activity-Dependent Presynaptic Glutamate Release in the Medial Prefrontal Cortex of Vulnerable Rats and Ketamine Rescues This Change

Adult male rats were exposed to CMS for 5 weeks, deemed CMS-R or CMS-V based on the sucrose preference test, and CMS-V animals were injected with subanesthetic ketamine (10 mg/kg) 24 h before sacrifice; ketamine completely restored depressive-like behavior. The behavioral characterization of the animals used in the present study is published in [15,17].

First, we measured the presynaptic release of glutamate and GABA in purified mPFC synaptic terminals (synaptosomes) in superfusion (Figure 1).

We found no significant changes with respect to basal glutamate release (Figure 1A), while significant differences were observed in depolarization-evoked glutamate release (Figure 1B). In particular, the depolarization-evoked glutamate release increased in CMS-V rats compared to CMS-R (with a trend for increase also compared to the controls) (Tukey’s post hoc tests: CMS-V vs. CMS-R *p* < 0.001, CMS-V vs. CNT *p* = 0.0607). Of note, ketamine restored glutamate release to CMS-R levels (Tukey’s post hoc test: CMS-V+KET vs. CMS-V *p* < 0.01). Both basal- and depolarization-evoked GABA releases were not significantly different among the experimental groups (Figure 1C,D).

### 2.2. Ketamine Increases Glucocorticoid Receptor Phosphorylation in the Medial Prefrontal Cortex of Vulnerable Rats

We have previously shown that CMS induces a long-lasting activation of the HPA axis, as demonstrated by increased serum corticosterone levels, which were higher in CMS-V than in CMS-R and unaffected by acute ketamine [15]. Here, we asked whether the protein levels of mineralocorticoid receptors (MRs) and glucocorticoid receptors (GRs) and GR activation by phosphorylation were altered in the total mPFC homogenates of rats exposed to CMS and ketamine treatments. We found no significant changes in MR and GR expression (Figure 2A,B). Nevertheless, ketamine significantly increased the phosphorylation of GR at Ser^232^ compared to both controls and CMS-V rats (Figure 2C) (Dunn’s post hoc tests: CMS-V+KET vs. CNT *p* < 0.05, CMS-V+KET vs. CMS-V *p* < 0.05).

### 2.3. Modulation of AMPA, NMDA, and mGluR2 Glutamate Receptors Induced by Stress and Ketamine in the Medial Prefrontal Cortex of Rats

We assessed whether the alterations observed in presynaptic glutamate release were associated with changes in glutamate receptors. We measured the protein expression levels of NMDA receptor subunits, expression, and phosphorylation levels of AMPA receptor subunits and the expression of mGluR2 in both total homogenate and synaptic membranes (containing postsynaptic density, PSD). Among mGluRs, mGluR2 was selected because we have previously found that CMS selectively induced a decrease in mGluR2 in vulnerable rats in the hippocampus [16] and because of interest in using mGluR2 as a putative target for the development of novel antidepressants [20,21].

In the total mPFC homogenate (Figure 3), no changes in NMDA or AMPA subunit expression and phosphorylation were found except for an increase in GluN2A levels in CMS-V+KET rats compared to controls (Figure 3B) (Tukey’s post hoc test: CMS-V+KET vs. CNT *p* < 0.05).

On the other hand, mGluR2 expression was significantly decreased in CMS-V rats, and ketamine reverted this change (Figure 3I) (Dunn’s post hoc test: CMS-V vs. CNT *p* < 0.05, CMS-V vs. CMS-R *p* < 0.05, CMS-V+KET vs. CMS-V *p* < 0.001).

More changes were found in synaptic membranes. The GluN1 NMDA receptor subunit levels increased in CMS-V+KET compared to CMS-V rats (Figure 4A) (Tukey’s post hoc test: CMS-V+KET vs. CMS-V *p* < 0.05), while no changes were found for GluN2A and GluN2B subunits (Figure 4B,C). Moreover, we found a significant decrease in GluA1 AMPA receptor subunit expression in CMS-V rats compared to the controls, which was fully recovered by ketamine (Figure 4D) (Tukey’s post hoc tests: CMS-V vs. CNT *p* < 0.05, CMS-V+KET vs. CMS-V *p* < 0.05). While no changes were detected with respect to GluA1 Ser^831^ phosphorylation, which increases the channel’s conductance [22] (Figure 4E), GluA1 phosphorylation at Ser^845^, which regulates the channel’s opening probability [22,23], increased selectively in CMS-R rats compared to both controls and CMS-V rats (Figure 4F) (Tukey’s post hoc test: CMS-R vs. CNT *p* < 0.05, CMS-R vs. CMS-V *p* < 0.05). Additionally, we observed a significant decrease in GluA2 expression in CMS-V+KET rats (Figure 4G) (Tukey’s post hoc test: CMS-V+KET vs. CNT *p* < 0.05) and in its phosphorylation at Ser^880^, which regulates receptor internalization [22] in CMS-R compared to the controls (Figure 4H) (Tukey’s post hoc test: CMS-R vs. CNT *p* < 0.05). Finally, similarly to the total homogenate, mGluR2 expression at synaptic membranes was reduced in CMS-V rats but not in CMS-R, although ketamine did not exert any effect (Figure 4I) (Tukey’s post hoc test: CMS-V vs. CNT *p* < 0.05, CMS-V vs. CMS-R *p* < 0.05).

### 2.4. GluA2 R/G Editing but Not Flip/Flop Splicing Is Reduced in the Medial Prefrontal Cortex of Stressed Rats and Rescued by Ketamine Treatment

The functional properties of the AMPA receptor subunit GluA2 are further regulated by epitranscriptomic changes and alternative splicing. Indeed, the GluA2 transcript (GRIA2) is subjected to RNA editing because adenosine deaminase acts on RNA (ADAR) enzymes, which comprises Arg/Gly (R/G) substitutions at an editing site located in the extracellular domain [24]. Moreover, the alternative splicing of two mutually exclusive exons can generate two GluA2 splicing isoforms, called “flip” and “flop” [24]. Notably, both these processes influence the kinetic properties of AMPA receptors because receptors containing edited R/G and “flip” variant GluA2 subunits recover faster from desensitization and have higher current responses to glutamate than receptors containing unedited subunits or the “flop” variant [24]. We measured whether CMS and ketamine could influence RNA editing at the R/G site and the alternative splicing of GluA2 [16] (Figure 5).

We found that CMS significantly reduced the R/G editing levels of the GluA2 transcript in both CMS-R and CMS-V rats, and ketamine recovered this change (Figure 5A) (Tukey’s post hoc tests: CMS-R vs. CNT *p* < 0.05; CMS-V vs. CNT *p* < 0.05; CMS-V+KET vs. CMS-V *p* < 0.05). No changes were found in the flip/flop RNA splicing ratio (Figure 5B).

### 2.5. Ketamine Rescues the Stress-Dependent Reduction in CaM Kinase II Phosphorylation but Not Bdnf mRNA Expression in the Medial Prefrontal Cortex of Rats

Finally, we evaluated two key molecular effectors of brain plasticity: the neurotrophin BDNF and calcium/calmodulin (CaM) kinase II. BDNF is the main neurotrophin in the adult brain, mediating neuronal development and synaptic function, and it is extensively studied for its central role in stress-related disorders and the mechanism of antidepressants [25,26,27]. We measured the total BDNF mRNA in the mPFC of our experimental groups, and as also previously observed in the hippocampus [15], we found a significant decrease in all stressed animals, independently of whether they were resilient, vulnerable, or treated with ketamine (Figure 6A) (Newman–Keuls post hoc tests: CMS-R vs. CNT *p* < 0.05; CMS-V vs. CNT *p* < 0.05; CMS-V+KET vs. CNT *p* < 0.05).

CaM kinase II is a Ser/Thr kinase accounting for 1–2% of the total brain protein, the most abundant protein in the PSD, and has critical roles in neurotransmitter release, synaptic plasticity, learning, and memory [28,29]. We measured its expression level and activating phosphorylation at the Thr^286^ residue in the total homogenate and synaptosomes. In the homogenate, the total CaM kinase II expression was unchanged (CNT 100 ± 3.568, CMS-R 92.77 ± 6.642, CMS-V 92.99 ± 2.353, CMS-V+KET 101.6 ± 5.754; one-way ANOVA, F(3,45) = 0.8861, *n* = 12). However, we found an effect of stress and ketamine on CaM kinase II phosphorylation at the Thr^286^ residue (Figure 6B). Indeed, we detected a significant decrease in Thr^286^ phosphorylation in CMS-V rats compared to both controls and CMS-R, and this change was completely reversed by ketamine (Tukey’s post hoc test: CMS-V vs. CNT *p* < 0.05; CMS-V vs. CMS-R *p* < 0.05; CMS-V+KET vs. CMS-V *p* < 0.05). A similar pattern was also present in mPFC synaptosomes. Indeed, while CaM kinase II expression was unchanged (CNT 100 ± 5.964, CMS-R 111.7 ± 5.508, CMS-V 98.44 ± 8.400, CMS-V+KET 101.3 ± 13.77; one-way ANOVA, F(3,19) = 0.3775, *n* = 8–10), the phosphorylation at Thr^286^ was again significantly decreased in CMS-V rats and rescued by ketamine (Figure 6C) (Tukey’s post hoc test: CMS-V vs. CNT *p* < 0.05; CMS-V+KET vs. CMS-V *p* < 0.05).

## 3. Discussion

In the present study, we found that chronic stress induces functional and molecular alterations in the mPFC of rats. Most changes were selectively observed in stress-vulnerable (anhedonic) animals and were recovered by acute subanesthetic ketamine, while others were found only in resilient animals or were induced by ketamine treatment (Table 1). Importantly, a few of these changes were also previously demonstrated in the hippocampus [15,16], while most are brain-area-specific.

Alterations selectively induced in behaviorally vulnerable animals and fully recovered by ketamine are particularly interesting because they could represent both the molecular underpinnings of depressive-like behavior and the possible targets of rapid antidepressant effects. We found that in the mPFC of CMS-V animals, ketamine normalized an increase in depolarization-evoked glutamate release and a decrease in mGluR2 expression in the homogenate, GluA1 in synaptic membranes, GRIA2 R/G editing, and CaM kinase II activation by phosphorylation. Intriguingly, these modifications, except for mGluR2 expression, appear to be brain-area-specific since they were not observed in the hippocampus of the same animals in previous studies [15,16].

As for presynaptic glutamate release, in the mPFC, we found no alterations of basal release but increased depolarization-evoked glutamate release in vulnerable rats, which was normalized by ketamine. Conversely, the changes observed in the hippocampus were different and partly opposite from those observed in the mPFC. Indeed, in the hippocampus, we reported decreased basal and depolarization-evoked release in vulnerable animals, while ketamine restored only basal glutamate release [15]. These findings, while they may seem conflicting, are in line with the hypothesis that ketamine, instead of inducing a general burst of glutamate release as generally believed [30,31,32], exerts a more complex homeostatic regulation of glutamate transmission, stabilizing glutamate dysfunction induced by stress [33,34]. Indeed, in brain areas where chronic stress induces a hypofunction of glutamatergic synapses, as in the hippocampus, ketamine restores glutamate release to control levels [15], while in the mPFC, where stress causes a glutamatergic hyperactivation, ketamine dampens glutamate efflux, thus stabilizing glutamatergic transmission. Accordingly, although most preclinical literature aimed at dissecting the modulatory effect of ketamine on glutamate transmission and at identifying molecular mechanisms of the fast antidepressant effect has been collected in naive animals [35,36,37,38,39], our data strongly suggest that the use of animal models of psychopathology is required to understand the mechanism of action of fast-acting antidepressants.

From a molecular point of view, the decrease in mGluR2 in stress-vulnerable animals suggests a reduction in presynaptic autoreceptors regulating glutamate release [21], and this is in line with the observed increase in the depolarization-evoked release of glutamate. On the other hand, the decrease in GRIA2 editing, GluA1, and CaM kinase II phosphorylation consistently suggests post-synaptic glutamatergic impairments in the mPFC of vulnerable animals. Indeed, GRIA2 editing controls AMPA receptor function since R/G-edited AMPA receptors have faster activation and desensitization kinetics than unedited subunits, thus controlling the excitability in response to subsequent stimuli [24,40]. At the same time, a reduction in GluA1 expression may be associated with the lower availability of this AMPA receptor subunit for the channel’s assembly, while a reduction in CaM kinase II activation may imply functional defects and the impairment of long-term potentiation (LTP).

Overall, our results suggest that chronic stress induces the dysregulation of glutamatergic synapses in the mPFC of vulnerable animals, exhibiting presynaptic hyperactivation but postsynaptic weakening, which could impact plasticity. This is in line with the previous literature showing that chronic stress impairs excitatory transmission and synaptic plasticity [41]. Importantly, acute ketamine rescued both presynaptic and postsynaptic alterations induced by stress, thus possibly normalizing glutamate transmission. Instead, as previously observed in the hippocampus, ketamine did not recover from the reduction in BDNF levels [15]. Nevertheless, here we only measured total BDNF mRNA expression, without considering that previous evidence proposed a role for local BDNF protein synthesis at synapses in the rapid antidepressant effect of ketamine [42].

Conversely, the increased GluA1 phosphorylation at Ser^845^ and decreased GluA2 phosphorylation at Ser^880^ in the mPFC synaptic membranes of resilient rats suggest that these mechanisms are implicated in the determination of a resilient phenotype. The phosphorylation of GluA1 at Ser^845^ increases the opening probability and peak amplitude of AMPA receptor currents and is involved in AMPA receptor trafficking relative to synaptic sites [43]. Therefore, the increased GluA1 phosphorylation at Ser^845^ in resilient animals suggests an enhancement of AMPA receptor function, which is the opposite of what happens in vulnerable rats. On the other hand, GluA2 phosphorylation at Ser^880^, regulating trafficking mechanisms, promotes the internalization of the receptor [43]. The permeability of AMPA receptors to calcium ions inversely depends on the presence of the GluA2 subunit within the tetramer, and most AMPA receptors contain GluA2 in the adult brain [44]. GluA2-lacking channels are difficult to detect; they are calcium-permeable, have faster decay kinetics but large conductance, and contribute to excitotoxicity [23,44]. Intriguingly, in the absence of the GluA2 subunit, AMPA receptors often consist of GluA1/GluA3 hetero-oligomers, which may lead to the reduced expression of the AMPA receptor at the synapse [45]. Thus, the concomitant decrease in GluA2 phosphorylation at Ser^880^ and increase in GluA1 phosphorylation at Ser^845^ could represent a protective mechanism against excitotoxicity and could be a way to preserve the physiological function of AMPA receptors. More studies, including electrophysiological recordings, are needed to unveil the functional changes induced by chronic stress in resilient animals and to understand to what extent these are required for the induction of a resilient phenotype. Of note, R/G GluA2 editing was reduced in resilient and vulnerable rats, suggesting that chronic mild stress may alter the activity of ADAR enzymes, as previously demonstrated with chronic social defeat in mice [46].

Finally, other molecular changes observed in the present study were reported only in stress-vulnerable animals after ketamine treatment, such as increased GR phosphorylation, GluN2A expression in the total homogenate and GluN1 in synaptic membranes, and reduced GluA2 expression in synaptic membranes. GR phosphorylation controls its activation, subcellular localization, transcriptional activity, and turnover [47], and it was reported to be increased also by chronic conventional antidepressants [48]. The regulation of the GR function by antidepressants has been proposed to play important roles in restoring physiological HPA axis activity [48].

The specific molecular changes in ionotropic glutamate receptor subunits in vulnerable animals treated with ketamine are in line with the increased glutamatergic transmission, since GluN1 is the core NMDA receptor subunit forming the channel pore, GluN2A is associated with enhanced excitatory postsynaptic currents, while reduced GluA2 levels could favor the assembly of calcium-permeable AMPA receptors [49].

In our study, we treated stress-vulnerable animals with ketamine only because the main aim was to study mechanisms of the rapid antidepressant effect, and as discussed above, the use of animal models of psychopathology is crucial in this context. Thus, we cannot say whether the alterations observed in ketamine-treated vulnerable animals are strictly implicated in rescuing the anhedonic phenotype. In any case, considering the modifications at mPFC glutamatergic synapses in vulnerable animals, the molecular changes in the ionotropic glutamate receptor subunits found in vulnerable animals treated with ketamine could contribute to restoring glutamatergic transmission. Again, electrophysiological studies are required to address this issue.

## 4. Materials and Methods

### 4.1. Animals

Experiments were performed in accordance with the European Community Council Directive 2010/63/UE and approved by the Italian legislation on animal experimentation (Decreto Legislativo 26/2014, authorization N 308/2015-PR). Adult Sprague–Dawley male rats (Charles River, Calco, Italy) were used. The rats were 175–200 g in weight at the beginning of the 5 weeks of the CMS protocol and 350–450 g at the end. Rats were housed two/three per cage at 20–22 °C, with a 12 h light–dark cycle (light on 7:00 a.m.; off 7:00 p.m.) and water and food ad libitum, except when required for CMS [15,17,18].

### 4.2. Chronic Mild Stress (CMS) Paradigm

CMS has been performed as previously reported [15,17,18]. Briefly, rats were exposed once or twice daily to random, mild, and unpredictable stressors (including food/water deprivation for 8–12 h, overcrowding for 6–12 h, isolation for 6–12 h, soiled cage for 6–12 h, cage tilting for 6–12 h, light-on overnight, light/dark reversal, and forced swim) for five weeks. CNT rats were undisturbed in their home cages except for behavioral testing (sucrose preference test) and weight measurement (twice a week).

### 4.3. Sucrose Preference Test

A sucrose preference test was used to assess the anhedonic phenotype and to divide stressed rats into CMS-resilient (CMS-R) and vulnerable (CMS-V) groups, as published previously [15,17,18]. For sucrose habituation (2 days before CMS start), rats were exposed to two bottles containing a 1% sucrose solution for 2 h, while for the sucrose preference test (one day before CMS, after three weeks, and 1 h before sacrifice), animals were exposed to two bottles, one containing 0.5% sucrose and one containing tap water, for 1 h. The position of the bottles was inverted after 30 min, and sucrose preference was calculated as follows: [sucrose solution intake (mL)/total fluid intake (mL)] × 100]. A cut-off of preference at 55% was adopted to deem the rats’ CMS-R or CMS-V [15] (Figure 7).

A total of 15 CNT, 15 CMS-R, 12 CMS-V, and 12 CMS-V+KET animals (from 2 independent sets) were used in the study.

### 4.4. Preparation of Subcellular Fractions

Immediately after sacrifice, mPFC was rapidly collected on ice, homogenized in 10 volumes of Tris-buffered 0.32 M sucrose, and centrifuged at 1000× *g* for 5 min to obtain the nuclear fraction [50]. The supernatant was gently stratified on a discontinuous Percoll^®^ gradient in Tris-buffered 0.32 M sucrose (2, 6, 10, and 20% (*v*/*v*)). After centrifugation (33,500× *g* for 5 min), purified synaptic terminals (synaptosomes) were collected at the layer between 10 and 20% Percoll and washed by centrifugation at 20,000× *g* for 15 min [15,34]. Synaptic membranes were prepared from synaptosomes by centrifugation, as reported in [51].

When used for neurotransmitter release experiments, synaptosomes were resuspended in a physiological medium: 140 mM NaCl, 3 mM KCl, 1.2 mM MgSO_4_, 1.2 mM CaCl_2_, 1.2 mM NaH_2_PO_4_, 5 mM NaHCO_3_, 10 mM glucose, and 10 mM HEPES, pH 7.4) [52,53]. For Western blotting experiments, homogenates, synaptosomes, and synaptic membranes were resuspended in a lysis buffer (120 mM NaCl, 20 mM HEPES, 0.1 mM EGTA, 0.1 mM DTT, phosphatase (Thermo Fisher Scientific, Milano, Italy), and protease (Sigma-Aldrich, Milano, Italy) inhibitors) [50].

### 4.5. Measurement of Neurotransmitter Release from Purified Synaptosomes

A total of 15 CNT, 15 CMS-R, 12 CMS-V, and 12 CMS-V+KET animals (from 2 independent sets) were used for neurotransmitter release experiments.

Synaptosomal aliquots were stratified on microporous filters at the bottom of 24 chambers thermostated at 37 °C (Superfusion System, Ugo Basile, Comerio, Italy; [54]). Synaptosomes were continuously superfused at a 0.5 mL/min speed for 48 min [55,56,57] and depolarized in superfusion (90 s pulse of 15 mM KCl-containing physiological medium) at *t* = 39 min. Three samples were collected as follows: two 3 min samples (*t* = 36–39 and 45–48 min; basal release) before and after and one 6 min sample (*t* = 39–45 min; stimulus-evoked release). Endogenous glutamate and GABA releases were quantified using high-performance liquid chromatography (HPLC), as previously described [58]. The stimulus-evoked overflow was estimated by subtracting the glutamate or GABA content of the two 3 min basal release samples from the content of the 6 min sample collected during and after the stimulating pulse.

### 4.6. Western Blotting

Samples for Western blotting were randomly selected among the animals used in release experiments. The number of animals used in each experiment is indicated in figure legends. A BCA concentration assay (Sigma-Aldrich) was used for protein dosage. Western blot analysis was carried out by incubating nitrocellulose or PVDF membranes (GE Healthcare Life Sciences, Piscataway, NJ, USA) containing electrophoresed and blotted proteins from the homogenate, presynaptic membranes, or synaptosomes [16,59]. The primary antibodies used were as follows: GluN1 (1:500, cod. AB9864, Millipore, Milano, Italy); GluN2A (1:500, AB1555P, Millipore); GluN2B (1:500, cod. 454,582, Millipore); GluA1 (1:200, cod. AGC-004, Alomone Labs, Jerusalem, Israel); GluA2 (1:2500, cod. AGC-005, Alomone Labs); phospho-pSer^831^ GluA1 (1:1000, cod. Ab109464, Abcam, Cambridge, UK); phospho-Ser^845^ GluA1 (1:1000, cod. ab3901, Abcam); phospho-Ser^880^ GluA2 (1:000, cod. Ab52180, Abcam); mGluR2 (1:1000, cod. ab15672, Abcam); MR (1:500, #sc-114112, Santa Cruz Biotechnology, Dallas, TX, USA); GR (1:500, #sc-1004, Santa Cruz Biotechnology); phospho-Ser^232^ GR (1:1000, #4161, Cell Signaling, Danvers, MA, USA); α-CaM kinase II (1:1000, #AB3111, Merck Life Science, Milano, Italy); phospho-Thr^286^ α-CaM kinase II (1:1000, #4161, Cell Signaling). Antibodies against GAPDH (1:6000, cod. Mab374, Merck Life Science), α-tubulin (1:20,000, cod. ab7291, Abcam), or β-Actin (1:20,000, #AB441, Sigma-Aldrich) were used as internal controls. Signal detection employed luminescence or chemiluminescence. Horseradish peroxidase (HRP)-conjugated anti-mouse and anti-rabbit antibodies (1:2000, Merck Life Science) or fluorophore-conjugated antibodies (1:2000, IRDye 800CW goat anti-rabbit IgG or IRDye 680RD goat anti-mouse IgG, LI-COR, Bad Homburg, Germany) were used as secondary antibodies. Chemiluminescent signals were detected using an enhanced chemiluminescence (ECL) kit (GE Healthcare Life Sciences) and visualized with Chemidoc XRS (Bio-Rad Laboratories, Hercules, CA, USA). Fluorescent signals were detected using an Odyssey infrared imaging system (LI-COR Biosciences, Bad Homburg, Germany) and analyzed using Odyssey version 1.1 (LI-COR Biosciences). Image Studio software version 5.2 (LI-COR Biosciences) was used for quantification.

### 4.7. RNA Isolation, Reverse Transcription, Real-Time PCR, RNA Editing, and Splicing Analysis

Samples for RNA isolation were randomly selected among the animals used in release experiments. Total RNA was extracted from rat mPFC using Tri-Reagent (Sigma-Aldrich, Milano, Italy) and Direct-zol RNA MiniPrep (Zymo Research, Freiburg, Germany), according to the manufacturer’s instructions. Reverse transcription was carried out using the iScript cDNA Reverse Transcription kit (BioRad Laboratories, Segrate, Italy). qPCR was performed using iTaq Universal SYBR Green supermix (Bio-Rad Laboratories). The primers used for qPCR were as follows: Bdnf For: GGCCCAACGAAGAAAACCAT; Bdnf Rev: CAGAAAGAGCAGAGGAGGCT. The relative expression of Bdnf was calculated using the comparative Ct (ΔΔCt) method and was expressed as a fold change [15]. The mean of P0 and SD18 (P0 For: AGTCGGAGGAATCCGATGAG, P0 Rev: ATTAAGCAGGCTGACTTGGTG; SD18 For: CATGCAGAACCCACGACAAT, SD18 Rev: CTTCCCATCCTTCACGTCCT) was used as a control reference.

The quantification for AMPA receptor subunit GRIA2 R/G editing levels was measured by sequence analysis as previously described (rG2-R/G-F: ATGAACGAGTACATCGAGCAGAGG, rG2-R/G-R: CCCCGACAAGGATGTAGAATACTC) [16,60,61]. Briefly, the electropherogram obtained after RT-PCR and sequencing analysis allowed calculating the editing level as a function of the ratio between the G peak area and A plus G peaks areas. The amount of each nucleotide was quantified using Discovery Studio (DS) Gene 1.5 (Accelrys Inc., San Diego, CA, USA).

The levels of the flip/flop splicing variants were evaluated by sequence analysis, as described previously [16]. Both GRIA2 isoforms were amplified using a pair of common primers (rG2-R/G-F: ATGAACGAGTACATCGAGCAGAGG; rG2-R/G-R: CCCCGACAAGGATGTAGAATACTC), and the flip and flop exon sequences appeared as peaks that were superimposed in the electropherogram. The relative expression level of the flip exon was calculated as the ratio of the peak area of G (flip) and the sum of peak areas A + G in different positions. The amount of each nucleotide was quantified using DS-Gene 1.5 (Accelrys Inc.).

### 4.8. Statistical Analysis

Statistical analyses of the data were performed using GraphPad Prism 9.5.1 (GraphPad Software Inc., San Diego, CA, USA). Normal distribution was verified using the Kolmogorov–Smirnov test. For normally distributed data, one-way analysis of variance (ANOVA) followed by Tukey’s post hoc multiple comparison tests was used. For non-normally distributed data, a Kruskal–Wallis test followed by Dunn’s multiple comparison tests was applied. Statistical significance was assumed at *p* < 0.05. Outliers were identified by applying the ROUT method (Q = 1%). Data were shown as the mean ± standard error of the mean (SEM).

## 5. Conclusions

In the present study, we extended our previous data collected in the hippocampus to the mPFC [15,16] in order to check functional and molecular alterations induced by stress and acute subanesthetic ketamine. We found that chronic mild stress has a remarkable impact on glutamatergic synapses in the mPFC of vulnerable animals, with signs of presynaptic hyperactivation and postsynaptic impairment, while protective mechanisms seem to be activated in resilient animals. On the other hand, acute ketamine normalized most alterations in vulnerable animals and activates mechanisms aimed at restoring glutamate transmission. Most changes were different from those previously observed in the hippocampus [15,16].

Although this study lacks electrophysiological measurements, which could give further information about the functional impact of the observed molecular changes, and did not consider possible sex differences, our results provide significant insights into the mechanisms of the rapid antidepressant effects of ketamine because it only involved male rats. This further supports the restorative action of ketamine against the maladaptive consequences of both acute and chronic stress [15,16,17,33,34,62,63,64].

## Figures and Tables

**Figure 1 ijms-24-10814-f001:**
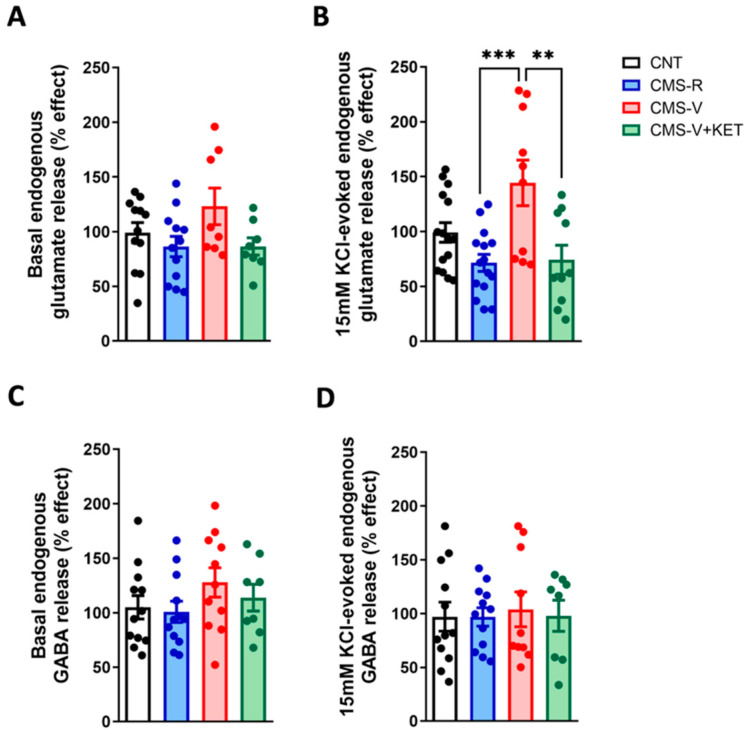
**Glutamate and GABA release from mPFC synaptosomes.** (**A**) Basal glutamate release from mPFC synaptosomes in superfusion; *n* = 8–12; one-way ANOVA, F(3,36) = 2.206, *p* > 0.05. The following is shown: (**B**) 15 mM KCl-evoked glutamate release from mPFC synaptosomes in superfusion; *n* = 10–15; one-way ANOVA, F(3,46) = 6.775, *p* < 0.001; Tukey’s post hoc tests: ** *p* < 0.01, *** *p* < 0.001. (**C**) Basal GABA release from mPFC synaptosomes in superfusion; *n* = 8–12; one-way ANOVA, F(3,39) = 1.126, *p* > 0.05. The following is shown: (**D**) 15 mM KCl-evoked GABA release from mPFC synaptosomes in superfusion; *n* = 8–12; Kruskal–Wallis test, H = 0.1118, *p* > 0.05.

**Figure 2 ijms-24-10814-f002:**
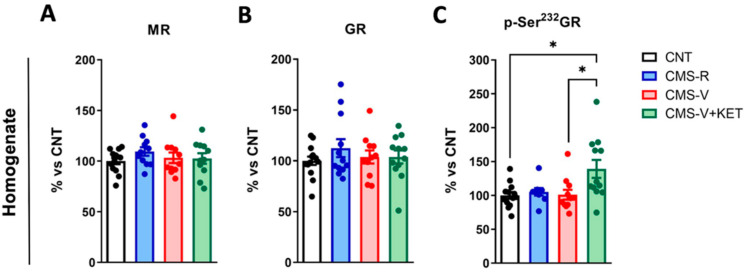
**MR and GR protein and phosphorylation in mPFC homogenate**. (**A**) MR protein expression levels in mPFC homogenates; *n* = 11–13; one-way ANOVA, F(3,42) = 0.8215, *p* > 0.05. (**B**) GR protein expression levels in mPFC homogenates; *n* = 11–14; one-way ANOVA, F(3,45) = 0.6731, *p* > 0.05. (**C**) Phospho-Ser^232^-GR/β-actin protein expression levels in mPFC homogenates; *n* = 11–14; Kruskal–Wallis test, H = 10.42, *p* < 0.05; Dunn’s post hoc test: * *p* < 0.05.

**Figure 3 ijms-24-10814-f003:**
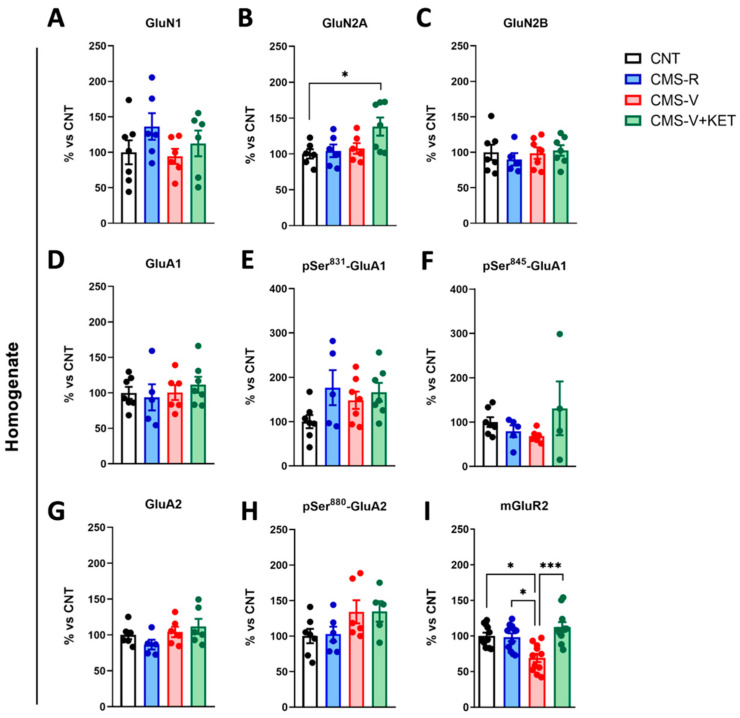
**Protein expression and phosphorylation of AMPA and NMDA receptor subunits and mGluR2 in mPFC homogenates.** Protein expression levels in the mPFC homogenates of (**A**) GluN1; *n* = 6–7; one-way ANOVA, F(3,21) = 1.259, *p* > 0.05, (**B**) GluN2A; *n* = 6–7; one-way ANOVA, F(3,21) = 3.563, *p* < 0.05; Tukey’s post hoc test: * *p* < 0.05, (**C**) GluN2B; *n* = 5–7; one-way ANOVA, F(3,22) = 0.3124, *p* > 0.05, (**D**) GluA1; *n* = 5–7; one-way ANOVA, F(3,21) = 0.3842, *p* > 0.05, (**E**) phospho-Ser^831^ GluA1; *n* = 5–7; one-way ANOVA, F(3,22) = 2.198, *p* > 0.05, (**F**) phospho-Ser^845^ GluA1; *n* = 4–7; Kruskal–Wallis test, H = 4.252, *p* > 0.05, (**G**) GluA2; *n* = 5–6; one-way ANOVA, F(3,19) = 1.684, *p* > 0.05, (**H**) phospho-Ser^880^ GluA2; *n* = 5–7; one-way ANOVA, F(3,20) = 2.217, *p* > 0.05, and (**I**) mGluR2; *n* = 11; Kruskal–Wallis test, H = 4.252, *p* < 0.001; Dunn’s post hoc test: * *p* < 0.05, *** *p* < 0.001.

**Figure 4 ijms-24-10814-f004:**
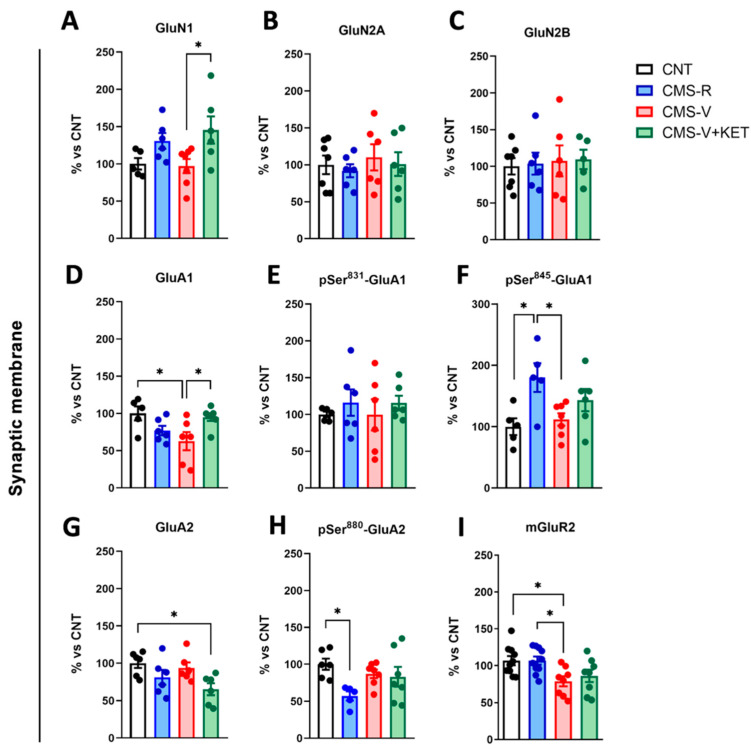
**Protein expression and phosphorylation of AMPA and NMDA receptor subunits and mGluR2 in mPFC synaptic membranes.** Protein expression levels in the mPFC synaptic membranes of (**A**) GluN1; *n* = 5–7; one-way ANOVA, F(3,20) = 3.701, *p* < 0.05; Tukey’s post hoc test: * *p* < 0.05, (**B**) GluN2A; *n* = 6–7; one-way ANOVA, F(3,21) = 0.2651, *p* > 0.05, (**C**) GluN2B; *n* = 5–7; one-way ANOVA, F(3,20) = 0.07126, *p* > 0.05, (**D**) GluA1; *n* = 5–7; one-way ANOVA, F(3,20) = 4.056, *p* < 0.05; Tukey’s post hoc test: * *p* < 0.05, (**E**) phospho-Ser^831^ GluA1; *n* = 5–6; one-way ANOVA, F(3,19) = 0.3577, *p* > 0.05, (**F**) phospho-Ser^845^ GluA1; *n* = 5–7; one-way ANOVA, F(3,19) = 4.465, *p* < 0.05; Tukey’s post hoc test: * *p* < 0.05, (**G**) GluA2; *n* = 6; one-way ANOVA, F(3,20) = 3.672, *p* < 0.05; Tukey’s post hoc test: * *p* < 0.05, (**H**) phospho-Ser^880^ GluA2; *n* = 5–7; one-way ANOVA, F(3,21) = 3.200, *p* < 0.05; Tukey’s post hoc test: * *p* < 0.05, and (**I**) mGluR2; *n* = 8–10; one-way ANOVA, F(3,32) = 4.630, *p* < 0.01; Tukey’s post hoc test: * *p* < 0.05.

**Figure 5 ijms-24-10814-f005:**
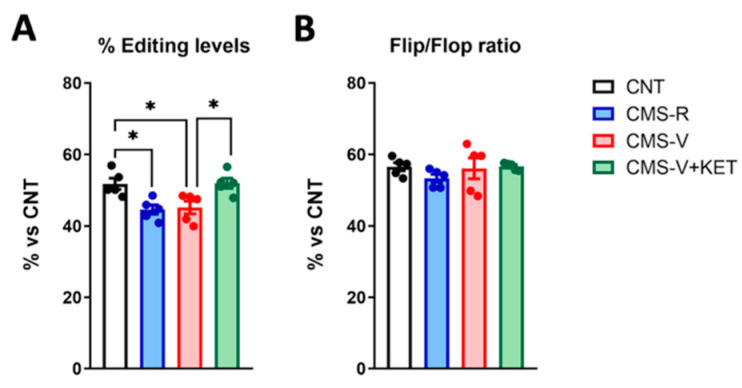
**AMPA GluA2 subunit RNA editing and splicing analysis.** (**A**) RNA editing levels at R/G site; *n* = 5; one-way ANOVA, F(3,16) = 7.084, *p* < 0.01; Tukey’s post hoc test: * *p* < 0.05. (**B**) Flip/flop splicing ratio of GluA2 mRNA; *n* = 5; one-way ANOVA, F(3,16) = 0.8825, *p* > 0.05.

**Figure 6 ijms-24-10814-f006:**
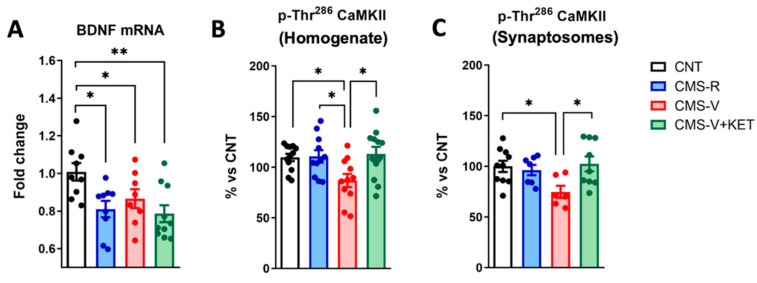
**CaM kinase II phosphorylation and BDNF mRNA expression in mPFC.** (**A**) Total BDNF transcript levels; *n* = 8–10; one-way ANOVA, F(3,32) = 4.724, *p* < 0.01; Newman–Keuls post hoc test: * *p* < 0.05, ** *p* < 0.01. (**B**) Phospho-Thr^286^-CaM kinase II (CaMKII)/β-actin protein expression levels in mPFC homogenates; *n* = 11–12; one-way ANOVA, F(3,42) = 4.055, *p* < 0.05; Tukey’s post hoc test: * *p* < 0.05. (**C**) Phospho-Thr^286^-CaMKII/β-actin protein expression levels in mPFC synaptosomes; *n* = 7–10; one-way ANOVA, F(3,28) = 3.365, *p* < 0.05; Tukey’s post hoc test: * *p* < 0.05.

**Figure 7 ijms-24-10814-f007:**
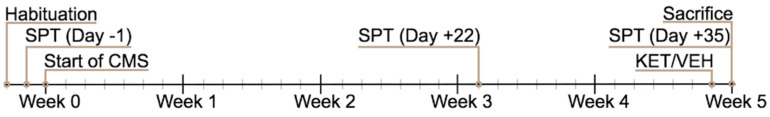
Experimental protocol timeline.

**Table 1 ijms-24-10814-t001:** Summary of functional and molecular differences between CMS-R, CMS-V, and CMS-V+KET animals in mPFC homogenates, synaptosomes, and synaptic membranes.

	CMS-R	CMS-V	CMS-V+KET
**Depolarization-evoked glutamate release**	=	= (↑ vs. CMS-R)	= (↓ vs. CMS-V)
**pSer^232^-GR** (homogenate)	=	=	↑
**GluN2A** (homogenate)	=	=	↑
**mGluR2** (homogenate)	=	↓	= (↑ vs. CMS-V)
**mGluR2** (synaptic membranes)	=	↓	=
**GluN1** (synaptic membranes)	=	=	= (↑ vs. CMS-V)
**GluA1** (synaptic membranes)	=	↓	= (↑ vs. CMS-V)
**pSer^845^-GluA1** (synaptic membranes)	↑	= (↓ vs. CMS-R)	=
**GluA2** (synaptic membranes)	=	=	↓
**pSer^880^-GluA2** (synaptic membranes)	↓	=	=
**pThr^286^-CaM kinase II** (homogenate and synaptosomes)	=	↓	= (↑ vs. CMS-V)

Arrows/equal signs outside the brackets represent the comparisons vs. control animals.

## Data Availability

Not applicable.

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
