# Peer review of "Functional and Molecular Changes in the Prefrontal Cortex of the Chronic Mild Stress Rat Model of Depression and Modulation by Acute Ketamine"

_ijms, 2023, doi:10.3390/ijms241310814_

Round 1

Reviewer 1 Report

The manuscript “Functional and molecular changes in the prefrontal cortex of the chronic mild stress rat model of depression and modulation by acute ketamine” by Mingardi and colleagues is a significant study on the molecular mechanisms underlying the fast-acting antidepressant effects of ketamine treatments. The use of a preclinical model of depression such as CMS-vulnerable rats makes this study particularly relevant. I found the results compelling and the descriptive table summarizing key results included in the discussion section very useful. However, I believe that the overall transparency of the presented data could be improved as follows:

The assignment of the animals to resilient and vulnerable groups is based only on the results of sucrose preference test. These results should be reported as they relevant data. I am aware of the drawbacks of such a lengthy protocol, but a more reasonable paradigm, in my opinion, should apply a z-score or principal component analysis to a battery of behavioral tests in order to evaluate overall resistance/vulnerability. Data would have been stronger if all the animals were treated with ketamine and a correlation between neurotransmitter/protein levels vs stress vulnerability would have been performed. 

I would expect that the same mice that went through the whole protocol would be studied in western blot for each protein analyzed. Why are there 11 mice/group for mGluR2 and fewer for the other proteins analyzed? N in each group is different across panels. Exclusion criteria should be at least stated in the methods section. Also, the original western blots for mGluR2, MR, GR, phospho-GR, CamKII, and phosho-CamKII are not attached to the manuscript.

Furthermore, discrepancies among the bands shown in the original western blots and the data reported in figure 3 are confusing: for example, the original western blot for GluN2A shows 32 samples, while in the data analysis, I can only see 6+6+6+7 data points (25 total) and the figure legend reports an n=5-7. Similar inconsistencies were observed for other panels in figure 3 and figure 4. Finally, the blot image of the homogenate pSer 880 GluA2 seems a different exposure of the total GluA2. The same issues are observed with other paired western blots (total/phospho).

mGluR3 shows higher expression levels than mGluR2 in the mPFC (according to data available on the Allen Brain Atlas and GTEX portal). Are mGluR3-specific antibodies available? If so, analysis of mGluR3 would be relevant to this study.

Total CamKII levels are described as unchanged but not shown.

p845-GluA1 (I assume this is the site p849) is a target of CamKII, therefore if these results suggest a direct link between CamKII activity and GluA1 phosphorylation in relation to the ketamine mechanism of action that should be reported to help the reader connect the dots.

Author Response

The manuscript “Functional and molecular changes in the prefrontal cortex of the chronic mild stress rat model of depression and modulation by acute ketamine” by Mingardi and colleagues is a significant study on the molecular mechanisms underlying the fast-acting antidepressant effects of ketamine treatments. The use of a preclinical model of depression such as CMS-vulnerable rats makes this study particularly relevant. I found the results compelling and the descriptive table summarizing key results included in the discussion section very useful.

We thank the reviewer for appreciating our work.

However, I believe that the overall transparency of the presented data could be improved as follows:

The assignment of the animals to resilient and vulnerable groups is based only on the results of sucrose preference test. These results should be reported as they relevant data. I am aware of the drawbacks of such a lengthy protocol, but a more reasonable paradigm, in my opinion, should apply a z-score or principal component analysis to a battery of behavioral tests in order to evaluate overall resistance/vulnerability.

We understand the point raised by the referee and agree that in principle using a battery of tests would give a more comprehensive behavioral characterization of the animals. However, as in previous studies by us (Tornese et al., Neurobiol Stress. 2019;10:100160; Elhussiny et al., Prog Neuropsychopharmacol Biol Psychiatry. 2021;104:110033; Mingardi et al., Neurobiol Stress. 2021;15:100381; Lamanna et al., Sci Rep. 2022;12(1):11055; Mingardi et al., Int J Mol Sci. 2023;24(2):1552) and others (Brivio et al., Transl Psychiatry. 2022;12(1):87; Gong et al., Front Mol Neurosci. 2021;14:633398; Papp et al., Psychopharmacology (Berl). 2022;239(7):2299-2307), behavioral changes were evaluated with sucrose preference test only. Indeed, being interested in the functional and molecular consequences of stress and ketamine treatment, we preferred to limit animal handling to a minimum. This especially in consideration of the fact that we used rats, which are more sensitive to multiple testing than mice. Moreover, SPT is the standard test for anhedonia used for deeming rats resilient and vulnerable in the CMS model of depression in rats (Papp and Willner Curr Protoc. 2023;3(3):e712; Strekalova et al., Psychopharmacology (Berl). 2022;239(3):663-693).

As regards the behavioral characterization of the animals used in the present study, we have not reported the data in the present ms because they have already been published (Tornese et al., Neurobiol Stress. 2019;10:100160; Elhussiny et al., Prog Neuropsychopharmacol Biol Psychiatry. 2021;104:110033). Thus, to avoid duplication, we have cited the published papers in the text (p. 3).

Data would have been stronger if all the animals were treated with ketamine and a correlation between neurotransmitter/protein levels vs stress vulnerability would have been performed. 

We understand this concern of the reviewer. Treating control and resilient animals with ketamine would exclude any nonspecific effects of the drug. However, this would have made a total of 6 experimental groups, with over 90 rats, difficult to reconcile with the current strict regulations of the Italian agency on animal welfare. Moreover, previous literature showed that while ketamine significantly increases sucrose consumption in mice after 28 days of CMS, no effect was seen in naive mice (Autry et al., Nature 2011 475:91-95).

Moreover, it is noteworthy that in the present work we did not look at behavioral/molecular changes in CMS rats taken as a whole but specifically analyzed these readouts in CMS-R and CMS-V rats. This makes sense if one considers that ketamine is administered to patients with MDD, very often treatment-resistant. 

As regards correlations, we don’t think they could be particularly informative in this context. Nevertheless, following the Referee’s suggestion, we tried correlating sucrose preference with depolarization-evoked glutamate release without finding any significant correlation (see below).

I would expect that the same mice that went through the whole protocol would be studied in western blot for each protein analyzed. Why are there 11 mice/group for mGluR2 and fewer for the other proteins analyzed? N in each group is different across panels. Exclusion criteria should be at least stated in the methods section. Also, the original western blots for mGluR2, MR, GR, phospho-GR, CamKII, and phosho-CamKII are not attached to the manuscript. 

We thank the reviewer for this comment and apologize for not clearly indicating that samples came from different experimental sets. At the same time, exclusion criteria were not mentioned in the original version of the ms.

A total of 15 CNT, 15 CMS-R, 12 CMS-V and 12 CMS-V+KET animals (from 2 independent sets) were used for neurotransmitter release experiments.

Western blotting and qPCR samples were randomly selected among animals used in release experiments. Two different research groups performed Western blotting experiments. One lab measured AMPA and NMDA subunits using 7 animals/group. Another lab measured corticosterone receptors, mGluR2 and CaMKII. 14 CNT, 12 CMS-R, 12 CMS-V and 12 CMS-V+KET animals/group were used for corticosterone receptors; 11 animals/group for mGluR2 in the homogenate; 10 CNT, 10 CMS-R, 8 CMS-V and 8 CMS-V+KET animals/group for mGluR2 in synaptic membranes; 12 animals/group for CaMKII in the homogenate; 10 CNT, 8 CMS-R, 8 CMS-V and 10 CMS-V+KET animals/group for CaMKII in synaptosomes.

In all experiments, outliers were identified by applying the ROUT method (Q=1%).

We understand that such heterogeneity in N could seem unjustified, but the reviewer should consider that Western blotting experiments were performed by different people, from different labs, over time.

More methodological details, including outlier identification, have been included in the revised document (see Methods).

We apologize for not having taken enough care to prepare supplementary material: missing blots are now shown.

Furthermore, discrepancies among the bands shown in the original western blots and the data reported in figure 3 are confusing: for example, the original western blot for GluN2A shows 32 samples, while in the data analysis, I can only see 6+6+6+7 data points (25 total) and the figure legend reports an n=5-7. Similar inconsistencies were observed for other panels in figure 3 and figure 4. Finally, the blot image of the homogenate pSer 880 GluA2 seems a different exposure of the total GluA2. The same issues are observed with other paired western blots (total/phospho).

We thank the reviewer for this comment and apologize for not having taken enough care to prepare captions which have now been corrected.

As for the number of bands in blots, discrepancies come from the fact that, since multiple gels were used to load all the samples, in some experiments, we used external standards, to provide further quality control. Moreover, some bands were excluded because they were identified as outliers.

Finally, as regards similarities between total/phospho proteins, this depends on the fact that the same membrane has been used to analyze the two forms, stripping the membrane after phospho-protein detection.

mGluR3 shows higher expression levels than mGluR2 in the mPFC (according to data available on the Allen Brain Atlas and GTEX portal). Are mGluR3-specific antibodies available? If so, analysis of mGluR3 would be relevant to this study.

We thank the reviewer for this suggestion. The study of mGluR3 would definitely be interesting. However, selective antibodies for mGluR3 with no cross-reactivity with mGluR2 have been only recently developed. Unfortunately, we do not have and never tested a mGluR3 selective antibody. Here, as also stated in the main document (p. 4), among mGluRs, mGluR2 was selected because we have previously found that CMS selectively induced a decrease of mGluR2 in vulnerable rats in the hippocampus (Elhussiny et al., Prog Neuropsychopharmacol Biol Psychiatry. 2021;104:110033) and because of its interest as a putative target for the development of novel antidepressants (Jiang et al., Cell. Mol. Neurobiol. 2022, ahead of print; Musazzi, Expert Opin. Drug Discov. 2021, 16, 147–157).

Investigation of  possible involvement of mGluR3 in future studies is warranted.

Total CamKII levels are described as unchanged but not shown.

Total CaMKII levels are given in the text (p. 8).

p845-GluA1 (I assume this is the site p849) is a target of CamKII, therefore if these results suggest a direct link between CamKII activity and GluA1 phosphorylation in relation to the ketamine mechanism of action that should be reported to help the reader connect the dots.

Different protein kinases can phosphorylate GluA1 at multiple sites (Lu and Roche Curr Opin Neurobiol. 2012;22(3):470-9). The phosphorylation at Ser845 is one of the most widely studied and characterized. Ser845 is a substrate for PKA and its phosphorylation enhances the open probability of homomeric GluA1 receptors (Roche et al., Neuron. 1996;16(6):1179-88). It has also been shown that antidepressants increase GluA1 phosphorylation at Ser845.

Conversely, Ser831 is a consensus site for both CaMKII and PKC (Lu and Roche Curr Opin Neurobiol. 2012;22(3):470-9). As we did not measure any significant changes in GluA1 phosphorylation at Ser831 either in the homogenate or at synaptic membranes, we deemed it appropriate not to discuss possible relationships between changes in CaMKII activation and GluA1 phosphorylation.

Reviewer 2 Report

1.Please provide the total numer of animals used in the study and the number of animals per group. In line with this, since the number of animals per group vary, what was the reason of such differences?

2. As it is hard to follow, please provide the diagram presenting point-by-point the designed behavioral study.

3. What was the reason to administer ketamine 24 h before the surgery? The half-life of ketamine is shorter, therefore why did the Authors not perform the surgery earlier?

4. The Authors stated that rats were exposed to CMS based on the sucrose preference test. I'm wondering whether rats behave similarly in the aspect of stress? Some test for depressive behavior or cognition should be done.

moderat English corrections are required

Author Response

1.Please provide the total number of animals used in the study and the number of animals per group. In line with this, since the number of animals per group vary, what was the reason of such differences?

We thank the reviewer for this comment and apologize for not clearly indicating that samples came from different experimental sets. At the same time, exclusion criteria were not mentioned in the original version of the ms.

A total of 15 CNT, 15 CMS-R, 12 CMS-V and 12 CMS-V+KET animals (from 2 independent sets) were used for neurotransmitter release experiments.

Western blotting and qPCR samples were randomly selected among animals used in release experiments. Two different research groups performed Western blotting experiments. One lab measured AMPA and NMDA subunits using 7 animals/group. Another lab measured corticosterone receptors, mGluR2 and CaMKII. 14 CNT, 12 CMS-R, 12 CMS-V and 12 CMS-V+KET animals/group were used for corticosterone receptors; 11 animals/group for mGluR2 in the homogenate; 10 CNT, 10 CMS-R, 8 CMS-V and 8 CMS-V+KET animals/group for mGluR2 in synaptic membranes; 12 animals/group for CaMKII in the homogenate; 10 CNT, 8 CMS-R, 8 CMS-V and 10 CMS-V+KET animals/group for CaMKII in synaptosomes.

In all experiments, outliers were identified by applying the ROUT method (Q=1%).

We understand that such heterogeneity in N could seem unjustified, but the reviewer should consider that Western blotting experiments were performed by different people, from different labs, over time.

More methodological details, including outlier identification, have been included in the revised document (see Methods).

  1. As it is hard to follow, please provide the diagram presenting point-by-point the designed behavioral study.

The behavioral characterization of the animals used in the present study was not reported in the present ms because it has already been published (Tornese et al., Neurobiol Stress. 2019;10:100160; Elhussiny et al., Prog Neuropsychopharmacol Biol Psychiatry. 2021;104:110033). Thus, to avoid duplication, we have cited the published papers in the text (p. 4). A more detailed description and a timeline of behavioral experiments have been added in the Methods.

  1. What was the reason to administer ketamine 24 h before the surgery? The half-life of ketamine is shorter, therefore why did the Authors not perform the surgery earlier?

The time point for drug administration has been based on previous literature studying the antidepressant properties of ketamine in animal models. Indeed, although the acute pharmacological effects of ketamine last only a few hours, its therapeutic effects persist for a long time. Notably, the antidepressant effects typically peak around 24 hours after a single dose of ketamine in both humans and animal models of depression (Berman et al., Biol Psychiatry. 2000 Feb 15;47(4):351-4; Zarate et al., Arch Gen Psychiatry. 2006;63(8):856-64; Fitzgerald et al., PLoS One. 2019;14(4):e0215554; Polis et al., Behav Brain Res. 2019;376:112153).

  1. The Authors stated that rats were exposed to CMS based on the sucrose preference test. I'm wondering whether rats behave similarly in the aspect of stress? Some test for depressive behavior or cognition should be done.

We agree that in principle using a battery of tests would give a more comprehensive behavioral characterization of the animals. However, as in previous studies by us (Tornese et al., Neurobiol Stress. 2019;10:100160; Elhussiny et al., Prog Neuropsychopharmacol Biol Psychiatry. 2021;104:110033; Mingardi et al., Neurobiol Stress. 2021;15:100381; Lamanna et al., Sci Rep. 2022;12(1):11055; Mingardi et al., Int J Mol Sci. 2023;24(2):1552) and others (Brivio et al., Transl Psychiatry. 2022;12(1):87; Gong et al., Front Mol Neurosci. 2021;14:633398; Papp et al., Psychopharmacology (Berl). 2022;239(7):2299-2307), behavioral changes were evaluated with sucrose preference test only. Indeed, being interested in the functional and molecular consequences of stress and ketamine treatment, we preferred to limit animal handling to a minimum. This especially in consideration of the fact that we used rats, which are more sensitive to multiple testing than mice. Moreover, SPT is the standard test for anhedonia used for deeming rats resilient and vulnerable in the CMS model of depression in rats (Papp and Willner Curr Protoc. 2023;3(3):e712; Strekalova et al., Psychopharmacology (Berl). 2022;239(3):663-693).

Round 2

Reviewer 1 Report

According to the authors’ response, the model used should be more appropriately defined as “stress-induced anhedonia”. I would suggest adjusting the title and text accordingly.

None of the literature suggested in the response reports that vulnerable animals were identified with multiple behavioral tests. I understand that rats are sensitive to multiple tests, but isn’t the core concept of the UCMS protocol to induce a depressive-like phenotype with multiple stressors? Are the authors implying that using a battery of behavioral tests would induce a depressive-like behavior in the resilient rats? This would be worrisome in the interpretations of this field of research.

Many western blots shown are saturated (i.e. pCamkII), burnt (i.e. actin), or show multiple bands (i.e. MR and GR). Each of these elements could affect the quantification of protein expression and, as a consequence, data interpretation. To enhance clarity, please provide at least images with a shorter exposure time and specify which bands were quantified when multiple bands are observed.

Author Response

According to the authors’ response, the model used should be more appropriately defined as “stress-induced anhedonia”. I would suggest adjusting the title and text accordingly.

None of the literature suggested in the response reports that vulnerable animals were identified with multiple behavioral tests. I understand that rats are sensitive to multiple tests, but isn’t the core concept of the UCMS protocol to induce a depressive-like phenotype with multiple stressors? Are the authors implying that using a battery of behavioral tests would induce a depressive-like behavior in the resilient rats? This would be worrisome in the interpretations of this field of research.

We thank the reviewer for the through revision of or ms.

As he/she recognizes, all the papers mentioned in the response report applied exclusively the sucrose test for deeming the animals vulnerable or resilient to stress. Being that papers published in different journals and by different groups (including the Paul Willner group, who originally established the model), we believe that this gives enough background to justify our approach. Moreover, the use of the term “stress induced anhedonia” would be confounding for the reader, since if anyone familiar with the literature on CMS can understand what “chronic mild stress rat model of depression” stands for, the same can’t be said for “stress induced anhedonia”. Finally, the ms. contains all the methodological details required to understand how the model has been built, thus the reader can reason about the limitations of the model.

Many western blots shown are saturated (i.e. pCamkII), burnt (i.e. actin), or show multiple bands (i.e. MR and GR). Each of these elements could affect the quantification of protein expression and, as a consequence, data interpretation. To enhance clarity, please provide at least images with a shorter exposure time and specify which bands were quantified when multiple bands are observed.

As the reviewer certainly knows, multiple expositions are required for WB studies and quantification software allows for image optimization. As for multiple bands in the same membrane, unfortunately this is a relatively frequent problem. For the identification of the band, we referred to the face sheet of the antibody and previous published literature. For the identification of GR, we superimposed the signal to that obtained for pGR in the same membrane. Images have been updated accordingly.

Reviewer 2 Report

The Authors have now provided sufficient explanation to the questions / sugestions mare by the Reviewer. Therefore it can be published in the form presented

It is ok

Author Response

We thank the reviewer for the through revision of or ms. and for having approved publication